# Characterization and Future Distribution Prospects of “*Carciofo di Malegno*” Landrace for Its In Situ Conservation

**DOI:** 10.3390/plants13050680

**Published:** 2024-02-28

**Authors:** Davide Pedrali, Marco Zuccolo, Luca Giupponi, Stefano Sala, Annamaria Giorgi

**Affiliations:** Centre of Applied Studies for the Sustainable Management and Protection of Mountain Areas-CRC Ge.S.Di.Mont., University of Milan, 25048 Edolo, Italy; davide.pedrali@unimi.it (D.P.); marco.zuccolo@unimi.it (M.Z.); stefano.sala1@unimi.it (S.S.); anna.giorgi@unimi.it (A.G.)

**Keywords:** agro-biodiversity, *Cynara cardunculus* subsp. *scolymus*, outline analysis, Camonica Valley, chlorogenic acid, cynarine, species distribution models, MaxEnt, Southern Alps

## Abstract

“Carciofo di Malegno” is a little-known landrace of *Cynara cardunculus* subsp. *scolymus* cultivated in Camonica Valley (northern Italy). The morphological and phytochemical characteristics of this landrace were investigated; furthermore, a species distribution model (MaxEnt algorithm) was used to explore its ecological niche and the geographical area where it could be grown in the future. Due to its spiky shape, “Carciofo di Malegno” was distinct from any other artichoke sample considered, and it appears to be similar to those belonging to the “Spinosi” group. The concentration of chlorogenic acid (497.2 ± 116.0 mg/100 g DW) and cynarine (7.4 ± 1.2 mg/100 g DW) in “Carciofo di Malegno” was comparable to that of the commercial cultivars. In “Carciofo di Malegno,” luteolin was detected in a significant amount (9.4 ± 1.5 mg/100 g DW) only in the stems and in the edible parts of the capitula. A MaxEnt distribution model showed that in the coming decades (2040–2060s), the cultivation of this landrace could expand to the pre-Alps and Alps of Lombardy. Climate change may promote the diffusion of “Carciofo di Malegno”, contributing to preservation and the enhancement of this landrace and generating sustainable income opportunities in mountain areas through exploring new food or medicinal applications.

## 1. Introduction

Nowadays there has been a significant loss of global agrobiodiversity with an estimated 75% decline [1,2]. Plant agrobiodiversity includes wild relatives, landraces, and modern cultivars of agricultural and food interest. Landraces are traditional crop varieties, locally adapted, that can hold “immense” value in terms of their agronomic and phytochemical-nutritional characteristics, as well as their ability to adapt to climatic change [3,4]. Many landraces exhibit unique profiles of secondary metabolites, antioxidants, and micronutrients, making them valuable resources for crop improvement and human health, but only a few of them have been conserved in situ (on-farm) and/or ex situ (in germplasm banks) [5,6,7,8,9].

The European Union (EU) has taken action to preserve agrobiodiversity with international strategies such as the EU Biodiversity Strategy 2020 [10], the National Recovery and Resilience Plan (PNRR) [11], and the 2030 Agenda for Sustainable Development [12]. These policies place the emphasis on the need to adopt innovative and sustainable solutions including the conservation and use of traditional cultivars and encouraging circular economy forms to invert the trend of genetic erosion in agriculture.

Italy has recognized the importance of preserving its agrobiodiversity and has taken various actions to promote the conservation of landraces both in situ (on-farm) and ex situ. In particular, Italy has established the National Register of Agrobiodiversity (Ministerial Decree 2019/39407; https://rica.crea.gov.it/APP/anb/ accessed on 1 September 2023) with the objective of protecting and promoting agricultural and food resources from the risk of extinction and genetic erosion. This register was established in December 2019 in accordance with Law No. 194/2015 (“Provisions for the conservation and enhancement of biodiversity of agricultural and food interest”). It operates under the supervision of the Ministry of Agriculture and Forestry Policies and is currently being set up and implemented. To register a landrace in the National Register of Agrobiodiversity, it is necessary to submit a series of documents to the relevant regional offices. These documents include the morphological description of the plant genetic resource, historical documentation proving its connection to the traditional cultivation area, a germplasm bank that conserves the seeds (or other propagation material) ex situ, and a list of farmers to preserve/cultivate it in its agroecosystem (on-farm conservation).

To implement conservation strategies on farms, it is essential to involve and incentivize “custodian farmers” who are responsible for in-purity seed production in the geographical area where landraces are traditionally cultivated. One strategy to engage custodian farmers, without relying on state economic subsidies, is to help them understand that cultivating/conserving and commercializing landraces can generate income [10]. In fact, some landraces possess phytochemical and nutritional characteristics of high value as well as exclusive sensorial aspects [5,13], making them highly sought after by consumers; this makes them valuable raw materials for the food and herbal industries. Unfortunately, only a few landraces have been thoroughly characterized [7,14]. This task is crucial for increasing knowledge about plant agrobiodiversity, promoting in situ conservation of landraces, and creating agri-food supply chains that foster sustainable development of territories [4].

For the same purpose, it would also be necessary to analyze the ecological niches of landraces, especially those that are propagated vegetatively, in order to understand where they could be cultivated/preserved in the near future based on environmental modifications due to climate change. Indeed, many cultivated plants are propagated vegetatively by humans (such as potatoes, garlic, asparagus, and artichoke), without producing and using seeds [15]. This prevents the evolution of populations and consequently hinders adaptation to the rapidly changing climate [13,16].

Italy is rich in agrobiodiversity and has a significant number of landraces, especially in mountainous and hilly areas [7,14]. According to recent census data [7], Italy has 40 landraces of artichoke (*Cynara cardunculus* L.; family: Asteraceae; chorotype: steno-mediterranean, [17]) cultivated on farms, of which only one (“*Carciofo di Malegno*”) is traditionally cultivated in the Southern Alps outside the Mediterranean basin (Figure 1). In the territory of the municipality of Malegno (Camonica Valley, Brescia Province, Lombardy region; latitude 45°57′06″N; longitude 10°16′30″E), a traditional cultivar of artichoke called “*Carciofo di Malegno*” is grown, and its flower heads were already appreciated in the early 20th century [18]. The inhabitants of Malegno have cultivated this landrace for decades, passing down their knowledge and expertise from one generation to the next. The local farmers declare that the “*Carciofo di Malegno*” is tastier, more bitter, and smaller than the artichoke commercial varieties [personal communication]. This landrace is traditionally propagated vegetatively by collecting and planting (in autumn or spring) the suckers produced by the plants [personal communication]. The “*Carciofo di Malegno*” exhibits the morphological characteristics of a globe artichoke of *Cynara cardunculus* L. subsp. *scolymus* (L.) Hayek (identified by the authors using “Flora d’Italia” dichotomous keys) [17].

The globe artichoke is considered an excellent source of fiber and minerals but also receives a place in folk medicine as a traditional herbal remedy for its beneficial properties, including hepatoprotective, choleretic, diuretic, and lipid-lowering activities [16,19]. Indeed, apart from its nutritional values, globe artichoke is characterized by a high content of polyphenols that are considered strictly related to its recognized healthy properties [19,20].

These compounds provide many health benefits in human life, including antioxidant, anti-inflammatory, cardio-protective, anticancer, anti-aging, and antimicrobial activities [21,22], and it is recognized that a diet rich in polyphenols may be associated with a reduced risk of chronic diseases as diabetes and cardiovascular, cerebrovascular, and neurodegenerative diseases [22,23]. In the globe artichoke, the main represented polyphenols are the compounds belonging to the family of caffeoylquinic acids (CQAs) together with *O*-glycosylated flavonoids like apigenin and luteolin derivatives [24] Several mono- and di-caffeoylquinic acids derivatives have been identified in globe artichoke extracts, and, among them, chlorogenic acid (5-*O*-caffeoylquinic acid) is the major caffeoylquinic acid derivative. Conversely, cynarin (1,3-*O*-dicaffeoylquinic acid), despite not being the most abundant, is the better-known derivative of this class of globe artichoke [19]. Cynarin has been identified both in capitula and leaves extracts and has been recognized as the compound responsible for the choleretic and cholesterol-lowering activities of the globe artichoke [19,25]. Regarding the “*Carciofo di Malegno*”, no studies have been conducted to characterize it (as it has never been involved in protection and valorization programs). It would be interesting to describe this landrace morphologically, ethnobotanically, ecologically, and phytochemically as well as discover its ecological niche to comprehend its potential present and future cultivation/conservation sites.

This research aims to characterize “*Carciofo di Malegno*” from a morphological perspective (essential for its registration in the National Register of Agrobiodiversity) and its phytochemical properties (essential for farmers and consumers). In particular, the morphological aspect was investigated by morphometric analysis while the phytochemical composition was analyzed using HPLC equipment. Furthermore, the study aims to determine the current and future geographical areas where the “*Carciofo di Malegno*” can be grown in the Southern Alps, providing tools (maps of the ecological niche modeling) that can be useful for its in situ conservation for those involved in actions for agrobiodiversity protection and sustainable development of the territory.

## 2. Materials and Methods

### 2.1. Plant Material

Figure 2 shows the five different globe artichoke genotypes included in this study: the “*Carciofo di Malegno*” (A) and four commercial globe artichokes (B, C, D, and E).

The “*Carciofo di Malegno*” (A) was collected from a local farmer, Felice Pezzoni, growing it in the Commonality of Malegno (Camonica Valley; latitude 45°57′06″ N; longitude 10°16′30″ E), who reported cultivating it from the 40s. The capitula for the analysis were collected from the plants in the field. Twenty capitula were collected manually with about 10 cm of stem. The four commercial globe artichokes (B–E) were collected from an Ortofrutta La Sicilia spadafora local shop that sells a different variety of Italian artichoke; this shop receives fresh artichokes every day. Genotype B was a “Romaneschi” type artichoke [26] and was originally grown in Sicily while both artichokes C and D belonged to the “Spinosi” group [26] and came, respectively, from Sardinia and Sicily. Artichoke E belonged to the “Catanesi” group [26] and came from Apulia. For each artichoke, 20 capitula were collected, and the stems were cut at about 10 cm of length.

In the laboratory, the capitula of all the artichokes were washed, examined to eliminate the damaged samples, and weighted (XB A220, Precisa Gravimetrics AG, Dietikon, Switzerland). Twenty capitula for each genotype were randomly selected for the morphometric analysis. The remaining were divided into three parts: non-edible flower head parts, stem, and receptacle with the inner bracts. The stems were cut at about 0.5–1 cm below the receptacle, weighted (Table 1, Figure 2), and subsequently cut in 5–6 parts. The capitula were turned removing the outer bract together with the upper spiny part of the flower head. The outer bracts were removed up to the fifth turn for the four commercial globe artichokes (B–E) and up to the third turn for the “*Carciofo di Malegno*” (A). The outer bracts and the upper spiny parts were weighted together as non-edible flower head parts.

Each part was frozen at −80 °C using a freezer (Haier Biomedical, Qingdao, P.R. China) and then milled with a kitchen mixer (Moulinex) to obtain a coarse powder. The powder was stored in a sealed container (50 mL, centrifuge tube) at −80 °C before the extraction.

### 2.2. Morphometric Analysis

Thirty samples for “*Carciofo di Malegno*” (A) and 30 artichokes for the commercial variety (B–E) randomly collected from local markets were used for the outline analysis (elliptical Fourier descriptors analysis) [27]. Each sample was placed on a white support table and photographed using a Canon EOS 2000D digital camera positioned perpendicular to the support surface. The images of the artichokes were processed using Adobe Photoshop CC2017 software: the shadows were removed, and the images were transformed to black and white. The outline coordinates were extracted with Momocs 1.4.0 [28,29] in an R environment [30] and converted into Fourier coefficients considering 9 and 15 harmonics (bracts and capitula, respectively) that gathered at least 99% of the total harmonic power [28]. In order to control left/right asymmetry, the artichokes were flipped in the same direction (considering as apex the most acute part of the artichoke), and then a landmark was defined at their base as a starting point for importing outline coordinates. The contours were centered, and the outline analysis was performed without numerical normalization.

Principal component analysis (PCA) was conducted on the matrix of coefficients to reduce dimensionality, and the samples were plotted on the first two axes (principal components). Linear discriminant analysis of principal components (DAPC) [31] was performed. The mean shape of the artichokes of each genotype was reconstructed using the MSHAPES function of Momocs, and multivariate analysis of variance (MANOVA) was performed to evaluate the differences in artichoke shape (bracts and capitula) among the five cultivars. Finally, pairwise MANOVA was used to highlight the differences between artichokes.

### 2.3. Phytochemical Analysis

All the solvents, reagents, and analytical standards were purchased from Merck (Milan, Italy).

The extracts for the HPLC assays were obtained by a two-step solvent extraction procedure adapting literature procedure [16,20]. An exactly weighed sample (1.5 g) of frozen artichoke powder was transferred in a screw-top cap conical-bottom centrifuge tube and extracted twice with methanol (1:10 g/mL *w*/*v* ratio) using an ultrasonic bath Digital Ultrasonic Cleaner MH020S (Vevor, USA) for 30 min at room temperature [20,24]. The supernatants were collected by centrifugation (Hermle z300, HERMLE Labortechnik GmbH, Wehingen, Germany) at 4000 rpm for 10 min. The residue was then extracted twice with 70% ethanol (1:10 g/mL *w*/*v* ratio) and sonicated for 30 min at room temperature. The collected supernatants were pooled together and evaporated to dryness under reduced pressure (rotary evaporator, LABOROTA 4000eco, Heidolph Instruments GmbH & Co., Schwabach, Germany) at 45 °C. The residue was re-dissolved in 60% methanol to a final volume of 5 mL. The resulting phenolic extract was stored at −20 °C overnight and then centrifuged at 4000 rpm for 10 min. A 1 mL aliquot of the extract was filtered through a nylon 0.2 µm Millex^®^-GN syringe filter prior to the chromatographic analysis.

An aliquot of the phenolic extract was subjected to acidic hydrolysis following the procedure developed by Nuutila et al. [32], with minor modifications. Briefly, 480 µL aliquot of the phenolic extract was mixed with 120 µL of 6 N hydrochloric acid and heated in a screw-top capped tube at 80 °C for 2 h. After cooling to room temperature, the mixture was diluted with 400 µL of methanol, sonicated, and filtered through a nylon 0.2 µm Millex^®^-GN syringe filter prior to the chromatographic analysis.

An aliquot of the phenolic extract was subjected to alkaline hydrolysis following the procedure developed by Lin et al. [33], with minor modifications. Briefly, 1 mL aliquot of the phenolic extract was taken to dryness under air flow. After that, the residue was dissolved with 300 µL of 4 N aqueous sodium hydroxide and incubated at room temperature for 18 h. After that, the mixture was acidified to pH by the addition of 150 µL of 12 N hydrochloric acid, diluted with 550 µL of methanol, and filtered through a nylon 0.2 µm Millex^®^-GN syringe filter prior to the chromatographic analysis.

The High-Performance Liquid Chromatography (HPLC) analysis was performed using a LC Agilent series 1200 apparatus (Waldbronn, Germany) consisting of a degasser, a quaternary gradient pump, an auto-sampler, and an MWD detector (Waldbronn, Germany). A Luna^®^ 5 μm C18 (150 × 4.6 mm) column (Phenomenex, Santa Clara, CA, USA) at 40 °C was used for this analysis. Sample injections were made at 10 µL for all samples and standards; the run time was 40 min. A binary gradient comprising 0.1% aqueous formic acid (*v*/*v*) (A) and acetonitrile (B) at a flow rate of 0.8 mL min^−1^ was used as the mobile phase. The gradient profile was as follows: 0 min, 5% B; 20 min, 25% B; and 30 min, 95% B; 40 min, 5%B. Absorbance wavelength was 330 ± 10 and 370 ± 10 nm, for the caffeoylquinic acids and for flavonoids, respectively.

The standards used were chlorogenic acid, cynarine, apigenin, and luteolin. Individual stock solutions of each standard were prepared using methanol at 1 mg/mL and stored at −20 °C. The working standard solutions were made by diluting the appropriate amount of each stock standard solution (1000 μg/mL) to obtain 5 calibration levels: chlorogenic acid, 25–800 μg/mL; cynarine, 0.5–6.25 μg/mL; caffeic acid, 12.4–400 μg/mL; apigenin, 1.042–50 μg/mL; and luteolin, 4.17–100 μg/mL.

Phytochemical data was analyzed using a one-way ANOVA test using R 3.2.1 software [30] to highlight the significant differences (*p* < 0.05) attributable to each genotype. All the results were expressed to dried weight of plant material.

### 2.4. Prediction of the Potential Distribution of “Carciofo di Malegno”

The fields where the “*Carciofo di Malegno*” is cultivated (and where it was cultivated in the latest decades) were identified by consulting the local farmers and were georeferenced using a GPS device (Garmin Etrex 32×). Twenty-five georeferenced points (occurrence points) were collected to assess the spatial distribution of “*Carciofo di Malegno*” in Malegno whose coordinates were imported into R in CSV format.

Nineteen bioclimatic variables (Table 2) were retrieved as predictors to model the potential environmental niche of “*Carciofo di Malegno*” based on its occurrence dataset. In particular, the bioclimatic layers were obtained from the World Climate Database (WorldClim 2.1, http://worldclim.org, accessed on 25 September 2023.) at a spatial resolution of 0.5 arc-s. All the bioclimatic variables were used to establish the distribution model of “*Carciofo di Malegno*” under the current conditions (2016–2020) and future global warming scenarios (2021–2040 and 2041–2060).

The latest iteration of climatic scenarios, used for the Sixth Phase of the Coupled Model Intercomparison Project (CMIP6, 2016–2021) and featured in the IPCC Sixth Assessment Report (AR6) [34], is based on a set of Shared Socio-economic Pathways (SSPs). The SSP-based scenarios are the most complex created to date and span a range from very ambitious mitigation (SSP1—sustainable development) to ongoing growth in emissions (SSP5—fossil-fueled development). The SSP2-4.5 scenario (“Middle of the road scenario”) was used in this research as it is an intermediate scenario compared to the two mentioned above. According to this scenario, the CO_2_ emissions will start to fall mid-century (without reaching net-zero by 2100), and the temperatures will rise 2.7 °C by the end of the century.

One global climate model (CNRM-CM6-1) was obtained from the WorldClim database for the future scenarios of the periods 2021–2040 and 2041–2060. The CNRM-CM6-1 is the recent fully coupled atmosphere-ocean general circulation model of the sixth generation jointly (developed by Centre National de Recherches Météorologiques and CMIP6) that replaced and improved the CNRM-CM5.1 model [33].

In this research, all models were run using the MaxEnt algorithm in R environment. The relative importance/weight of each bioclimatic preditor for the distribution model was assessed using the Jackknife test [35], and the response curves for each of the environmental variables were generated.

The accuracy of the resulting model was evaluated by computing the Area under the Curve (AUC) of the Receiver Operating characteristic Curve (ROC). AUC values range from 0 to 1, and the higher the value of AUC, the better the performance of the model.

The output of the MaxEnt application is a georeferenced raster file, indexing the environmental suitability of “*Carciofo di Malegno*” with values ranging from 0 (unsuitable) to 1 (optimal). All the raster files generated in this study were imported into QGIS 3.28 (http://qgis.osgeo.org) to produce the distribution maps of the “*Carciofo di Malegno*” for the current and future scenarios.

## 3. Results

### 3.1. Shape of the Flower Capitulum and Bracts

Figure 3a,b shows the PCA biplot of the Fourier coefficients calculated for the five artichokes’ capitula and the results of the Linear Discriminant Analysis of principal components (LDA), respectively. Along the first principal component (PC1 = 35.2%), the reconstructions of the capitula’s shape become rounder, while along the PC2 (22.6%), the apexes become more acute.

The samples were overlapped in the LDA biplot (Figure 3b) wherein sample B was positioned on the right part of the graph, separated from the other capitulum samples. In fact, Figure 3c displays the mean shape of the five artichoke samples and the “*Carciofo di Malegno*” was close to samples C and E while the artichoke B had a completely different form and was more rounded. These results were also confirmed by a MANOVA test that showed significant shape differences between the five artichokes’ capitula (F_96,145_ = 15.119, *p* < 0.01).

Figure 3d shows the PCA biplot of the artichokes’ bracts of the five artichokes’ genotypes where the first two principal components (PCs) explain 72% of total variance (PC1 = 49.2%; PC2 = 22.8%) although a linear discriminant analysis of principal components (LDA) biplot of the artichokes’ bracts is shown in Figure 3e. Both of these discriminant analyses showed that only artichoke B had a very diverse bracts shape among all samples. In the bottom left part of the PCA biplot, the bracts become more rounded, while along the second axis the apex becomes spikier.

Figure 3f displays the mean shape of bracts of each artichoke genotype. While the medium shape of samples A, C, D, and E is pointed, that of artichoke B is oval. The results of pairwise MANOVA (Figure 3f) confirmed the previous outcome and demonstrates that there are significant differences between the bracts shape of “*Carciofo di Malegno*” and the other four commercial varieties (F_52,145_ = 64.777, *p* < 0.01).

### 3.2. Phytochemical Characteristics

Figure 4a displays the chlorogenic acid content in the stems and in the edible and non-edible parts of the heads of the five cultivars included in the analysis. The content of chlorogenic acid resulted higher in the stems than in the edible and non-edible parts of heads for most of the selected artichokes. The stems of the artichoke B had the highest content (1414.0 ± 325.2 mg × 100 g^−1^ DW), followed by the artichoke C (1222.2 ± 196.2 mg × 100 g^−1^ DW) and the “*Carciofo di Malegno*” (A) (1032.9 ± 294.9 mg × 100 g^−1^ DW). Conversely, artichokes D and E showed a content in the stems significantly lower (76.0 ± 3.0 and 445.9 ± 22.6 mg × 100 g^−1^ DW, respectively) than that of the other cultivars. Moreover, the stems of the Sardinian artichoke (D) had a content of this caffeoylquinic acid lower than the edible and non-edible parts of heads, resulting in the sample with the lowest content. The chlorogenic acid content of the edible part of the heads ranged from 321.8 ± 126.0 mg × 100 g^−1^ DW and 814.8 ± 178.8 mg × 100 g^−1^ DW, resulting in higher than that of the non-edible parts of the heads. The heads of the “*Carciofo di Malegno*” (A) having a chlorogenic acid content of 497.2 ± 116.0 mg × 100 g^−1^ DW and 164.2 ± 42.7 mg × 100 g^−1^ DW in the edible and non-edible parts, respectively, were comparable with most of the other cultivars.

Figure 4b displays the cynarine contents of the samples. The heads of almost all the cultivars of artichoke showed a higher content of cynarine than the stems. In particular, this compound is concentrated in the edible parts showing a content ranging from 6.7 ± 0.4 mg × 100 g^−1^ DW to 8.2 ± 1.3 mg × 100 g^−1^ DW. Conversely, the non-edible parts had a lower content of cynarine with values approximatively 2-fold lower than the inner edible parts of the capitula. In artichokes C, D, and E, the stems had a lower cynarine content than the edible parts, while the stems “*Carciofo di Malegno*” (A) showed a similar content (7.4 ± 1.2 mg × 100 g^−1^ DW). The cynarine content of artichoke B stems (12.5 ± 0.7 mg × 100 g^−1^ DW) was higher than that of the capitula, resulting in the richest caffeoylquinic acid sample.

Figure 4c displays the free luteolin content measured after the acidic hydrolysis of sample extracts. The edible parts of the “*Carciofo di Malegno*” (A) resulted in one of the richest samples in free luteolin (9.4 ± 1.5 mg × 100 g^−1^ DW), showing a content similar to that of artichokes C and D (10.9 ± 4.0 and 9.8 ± 1.8 mg × 100 g^−1^ DW, respectively). Conversely, only a trace of luteolin was detected in the non-edible parts of A, while samples of the other cultivars resulted in higher content. In particular, the non-edible parts of artichoke D showed the highest luteolin content recorded (17.7 ± 2.0 mg × 100 g^−1^ DW). The luteolin was less concentrated in the stems than in the heads in most of the cultivars, apart from artichoke B. Artichoke E resulted in the poorest in luteolin, with only traces detected in all the parts.

Figure 4d displays the caffeic acid content measured after the basic hydrolysis of the stems, edible, and non-edible part extracts. The content of this compound in the samples followed the same distribution of the chlorogenic acid.

Figure 4e displays the free apigenin content measured after the acidic hydrolysis of sample extracts. Apigenin was more concentrated in the heads than in the stems for all the cultivars. Indeed, the stems of artichoke B showed a significant amount of free apigenin (3.3 ± 0.8 mg × 100 g^−1^ DW and 1.9 ± 0.0 mg × 100 g^−1^ DW, respectively) while in the others this flavonoid was detected only in traces. The capitula of “*Carciofo di Malegno*” had a higher content of apigenin, with results comparable to artichoke D and similar to artichoke C. In these cultivars, the apigenin was more concentrated in the inner part of the heads, while in artichokes B and E the external non-edible parts showed a higher content than the inner bracts and receptacle.

### 3.3. Current and Future Potential Distribution

Figure 5 shows the maps of the potential distribution of the “*Carciofo di Malegno* “ for the current and future scenarios. The predicted AUC value for the future periods was 0.96, indicating excellent predictions. Among the bioclimatic factors, temperature plays an important role in the definition of the ecological niche of “*Carciofo di Malegno*”. Isothermality (mean diurnal range/temperature annual range—BIO3) and the mean diurnal range [mean of monthly (max temp–min temp)—BIO2] are the topmost contributing factors accounting for 36% and 33.2% of the total contribution, respectively (Appendix A).

The *“Carciofo di Malegno*” has the potential to thrive in the medium-southern regions of the Camonica Valley, especially in its native zone (Malegno) and nearby areas including the valley floor of Camonica and Valtellina valleys situated at altitudes between 300 and 400 m above sea level. This landrace finds the appropriate, although not excellent, climatic conditions (probability of occurrence: 0.6–0.3) in the hilly areas of the Lombardy Prealps close to the Po Plain.

In the 2021–2040s period, the “*Carciofo di Malegno*” could be cultivated in other areas of the Camonica Valley (not only in Malegno) compared with the potential current distribution. In fact, the territory with a suitable bioclimate was significantly increased including Edolo as other municipalities in the upper part of the Camonica Valley, which are located at higher elevations than Malegno, altitude (600–1000 m asl). In the 2041–2060 scenario, the areas with favorable climatic conditions for the cultivation of the “*Carciofo di Malegno*” extend further north in Lombardy (and in Switzerland), while the probability of occurrence in the original area (Malegno) is low (0.4–0.2), and it is even more so in the hilly areas of the lower Camonica Valley and in the Po Plain (0.2–0.0).

## 4. Discussion

In this study, the artichoke outline analyses demonstrated that there are differences among the five genotypes assessed, particularly between “*Carciofo di Malegno*” and the commercial varieties.

Morphometric analysis showed that the landrace was different from each other artichoke sample. Due to its spiky shape, this landrace seems to be similar to the samples belonging to the “Spinosi” group (artichoke C and D) while was very different from sample B, which had a rounder profile typical of the “Romaneschi” category [26].

Modern geometric morphometric analyses (GMM), such as outline analysis, are an innovative way to better describe and compare shapes and forms. GMM methods find applications in numerous fields, spanning from plant biology to other disciplines. They enable the examination of the relative locations of landmarks and point sets utilized to approximate curves (outlines) and surfaces, facilitating the measurement of size and shape [29,36]. In recent years, botanists have also applied these types of assays to study the shape of leaves and other plant organisms. For example, Giupponi and Giorgi [27] were able to distinguish leaf shape between two varieties of *Primula albenensis*, while Chitwood and Otoni [37] analyzed more than 3300 leaves from 40 different *Passiflora* species using GMM. In 2020 Giupponi and his coworkers [8] used these methods to compare the shapes of three potato landraces (*Solanum tuberosum* L.) from the *Consorzio della Quarantina* (Genova, Italy) in order to find potential differences between different species and the comparison between populations of the same species for the definition of morphotypes and ecotypes. In particular, as previous work explored, geometric morphometric analyses are useful for the characterization and identification of landraces, and they could replace or supplement the morphological forms of variety description called Distinctness, Uniformity, and Stability (DUS) Testing [27].

GMM are effective and economical, and they allow an objective analysis minimizing human error; they only require plant samples and low-cost equipment like a digital scanner for image acquisition, software for digital analysis, and statistical analysis of data [27] Today, DUS testing redacted by the International Union for the Protection of New Varieties of Plants (UPOV) and the Community Plant Variety Office are mandatory to register cultivars both in the register of conservation varieties and in the register of plant agrobiodiversity. The possibility to have models of cultivar shapes uploaded and shared in an open tool (folder, dataset) to import in a GMM software could improve the comparison among landraces (or plants in general) and characterize new varieties accurately. Nowadays the standards present in DUS forms are drawn, and, in addition, not everyone has the opportunity to grow even the varieties of comparison to make a real contrast between cultivars.

Globe artichoke holds significance not only as a culinary vegetable but also as a medicinal plant and a source of secondary metabolites with health-promoting activities. This plant is indeed known for its abundant phenolic compounds, with caffeoylquinic acids being the primary constituents. Among these compounds, chlorogenic acid stands out as the most abundant derivative. However, it is cynarine that has garnered significant attention due to its beneficial and health-promoting properties [19].

In this study, the capitula of the “*Carciofo di Malegno*” showed a content of chlorogenic acid and cynarine comparable with that of the commercial cultivar. In all the artichokes included in the analysis, the receptacle and the inner fleshy bracts (edible parts) had a content of these two caffeoylquinic derivatives higher than the non-edible parts of the capitula. These observations agree with the literature-reported data [16,38]. The stems of artichokes A, B, and C had a higher content of chlorogenic acid compared to the heads, while, conversely, those of cultivars D and E showed significantly lower content. Possibly, this could be attributed to tissue aging. In fact, the stems of the latter artichokes were discovered to exhibit a greater degree of lignification. It is well established that the overall content of caffeoylquinic acids is closely linked to the physiological condition of the tissues, with a decrease in content observed in correlation with lignification [19,38]. In general, the stems of the analyzed cultivars demonstrated a lower cynarine content compared to the edible portions of the capitula. However, “*Carciofo di Malegno*” and sample C exhibited a similar content, while cultivar B displayed a higher content. These elevated levels signify the potential of these two cultivars in extracting valuable caffeoylquinic acids from artichoke stems, which are typically discarded as waste.

Apart from chlorogenic acid and cynarine, several other caffeoylquinic derivatives are present in the artichokes [19]. Hence, the total caffeic acid content was measured after basic hydrolysis as an indirect measure of the total caffeoylquinic acid. The content of caffeic acid followed a distribution in the different samples similar to that of chlorogenic acid. These results are unsurprising as chlorogenic acid is the most abundant caffeoylquinic derivative in artichoke, and, consequently this compound represents the main source of caffeic acid during the process of hydrolysis [39].

Luteolin and apigenin are the main flavonoids in globe artichoke and are present in the tissues as glycoside derivatives [39]. In this study, to assess the content of these two flavonoids, the extracts of the different samples were subjected to acid hydrolysis to free the aglycone from the saccharide moieties. In the “*Carciofo di Malegno*”, luteolin was detected in significant amounts only in the stems and in the edible parts of the capitula.

Conversely, a significative content of apigenin was detected only in the edible and non-edible parts of the capitula. The content of these two flavonoids in the different samples was comparable with that of the other cultivars.

These results suggest that the receptacle and inner bracts of the “*Carciofo di Malegno*” may represent an excellent dietary source of these two flavonoids. Moreover, the stems together with the outer non-edible parts of the capitula, which are typically considered as waste, may be used in herbal fields for the preparation of flavonoid-rich extracts in terms of circular economy.

From a phytochemical perspective, the “*Carciofo di Malegno*” has similar properties to the commercial artichokes and for this reason can be used for food (although this variety is smaller and therefore is not very competitive with others) and herbal use. These bioactive molecules are interesting for human health and can be used for the production of innovative-functional products too.

This information is essential to encourage farmers and land managers to create new local supply chains that allow the sustainable development of mountain areas and, at the same time, the in situ conservation of the landrace. In fact, much crop diversity is now held ex situ in gene banks or breeders’ materials rather than on-farm (in situ) [4].

To contrast this trend, the MaxEnt distribution model was combined with R language to predict the potential distribution of “*Carciofo di Malegno*” in the north of Italy under current and future climatic conditions.

Based on the analysis of the ecological niche of the Malegno artichoke, it emerged that it can currently be grown in a fairly restricted area in Lombardy (the valley floor of the Camonica and Valtellina valleys). However, considering climate change, it is probable that in the coming decades the area where this landrace could be cultivated will be much wider, including a large part of the pre-Alps and Alps of Lombardy (Figure 5).

In effect, due to the global warming that will affect the study area in the coming decades, the climatic optimum of the “*Carciofo di Malegno*” could be located at higher altitudes than the current ones. This scenario could make cultivation difficult (in situ conservation) of this landrace in the area where it is and has been cultivated up to now (Malegno), and therefore it is possible that the “*Carciofo di Malegno*” (as well as other landraces) could become a traditional variety of a territory and be cultivated outside that territory.

Any translocation of the “*Carciofo di Malegno*” outside the area where it is traditionally cultivated would create problems for its protection/enhancement through the current regulatory instruments for agrobiodiversity protection. In fact, both the European Register of Conservation Varieties [40] and the Italian Register of Agrobiodiversity protect the landraces and the geographical area where they are traditionally cultivated (and not the areas where they may be cultivated in the future). This problem/paradox could be solved by allowing a ten-year update of the registers and differentiating the area where landraces were traditionally cultivated (in the past) and the area (or areas) where is possible to cultivate them based on the contingent environmental conditions. 

It is very probable that in the future the “*Carciofo di Malegno*” will be able to be cultivated at higher altitudes than the current ones (since, according to the model used in this research, they will have the same climatic conditions) while it is less clear if in the future it will not be possible to grow it in the current areas (Malegno and the valley floor of Val Camonica in general) where the climate will be warmer.

Under the SSP145 scenario, horizon 2021–2040, the expansion of “*Carciofo di Malegno*” will increase, reaching other zones in the north of the Lombardy region, like high Camonica Valley, Valtellina, and other areas in the Alps. The tendency of this landrace to encroach hilly and mountain areas is presumably based on the temperature change that among climate factors plays an important role in the distribution of artichokes (Appendix A). However, in the 2041–2060s, the future potential habitat of “*Carciofo di Malegno*” under the SSP145 scenario seems to be less expanded than the previous temporal horizon. In fact, the temperature probably will be too high such that it will be impossible to grow artichokes in the original zone.

In fact, the plasticity (understood as the ability of plants to adjust their phenotype in response to different environmental conditions) [41] of this landrace (as well as that of most landraces) is not known; therefore its ability to grow and produce in different environmental conditions is not known. Ad hoc studies (ecological and/or physiological) could be conducted to clarify the real adaptive capacity of the “*Carciofo di Malegno*” as it would be interesting to understand, through genetic analyses, if the few custodian farmers have different populations of “*Carciofo di Malegno*” or the same genotype. Although the “*Carciofo di Malegno*” is traditionally propagated by agamic means (with the suckers), it is not improbable that there would be a genetic difference (even minimal) among the populations of the local farmers. If this were confirmed, cross-fertilization programs (among populations) and sexual propagation (with seeds) of the “*Carciofo di Malegno*” could be planned in order to minimize genetic erosion, improve genetic mixing, and promote evolution and adaptation of the landrace to environmental changes [42].

For example, Bonasia et al. [28] determined the total polyphenolic concentration and the antioxidant activity in five artichoke varieties of which three were hybrid. The latter showed a lower amount of polyphenols than the traditional cultivars, particularly in leaf waste. Mauromicale and his co-workers [43] carried out a two-year study wherein they shifted the harvest period of seed-grown globe artichokes on a new F1 seed-grown hybrid of artichoke [*Cynara cardunculus* L. var. *scolymus* (L.) Fiori] including “Violetto di Sicilia,” traditionally vegetatively propagated, as a comparison. The hybrid sample produced heads with low weight during late autumn—early winter; on the contrary, for “Violetto Siciliano” the effects of the studied factors were less evident.

The “*Carciofo di Malegno*” could also be an interesting genetic resource for commercial variety improvement programs [15,44]. In fact, the ability of the “*Carciofo di Malegno*” to grow (and complete its biological cycle) in an Alpine valley that has decidedly lower temperatures than those of the Mediterranean areas could be enhanced with the production of varieties/hybrids capable of growing in cold areas [16]. The latter could be used to extend the cultivation of artichokes outside the Mediterranean area in countries with cooler climates. For Italy, which is the most important producer of artichokes among European countries (more than 44,000 ha and 406,000 tons per year [39]), it would mean extending production to the mountain areas of the Apennines and the Alps.

## 5. Conclusions

This research has allowed us to characterize from the morphometric and phytochemical point of view the artichoke of Malegno from which it emerged that this landrace was distinct from any other artichoke sample due to its spiky shape, and it appears to be similar to those belonging to the “Spinosi” group because of its thorny shape. Modern geometric morphometric analyses better describe and compare shapes and forms, and they could integrate the classical morphological description of the varieties. The concentration of chlorogenic acid and cynarine in “*Carciofo di Malegno*” was comparable to that of the commercial cultivars. In “*Carciofo di Malegno*”, luteolin was detected in a significant amount only in the stems and in the edible parts of the capitula. In contrast, apigenin was detected only in the edible and non-edible parts of artichokes.

The area where the “*Carciofo di Malegno*” could be cultivated has been explored. GMM analysis has shown that in the coming decades (2040–2060s) it can be grown on a larger area than the current one, reaching another mountain environment (also outside Camonica Valley). The results of the characterization (morphological and phytochemical) and the analysis of the ecological niche of the “*Carciofo di Malegno*” will be useful to begin the registration practices to register it in the Italian Register of Agrobiodiversity, promoting events of dissemination of knowledge on the territory and therefore preserve this landrace in situ (on farm). This work is an example of how researchers and stakeholders should collaborate to encourage landraces in situ conservation, creating positive incentives for farmers and enterpreneurs to support territories that hold unique and little-known agri-food resources to establish a new supply chain (healthy agri-food and/or herbal). The latter could allow sustainable development of mountain areas.

## Figures and Tables

**Figure 1 plants-13-00680-f001:**
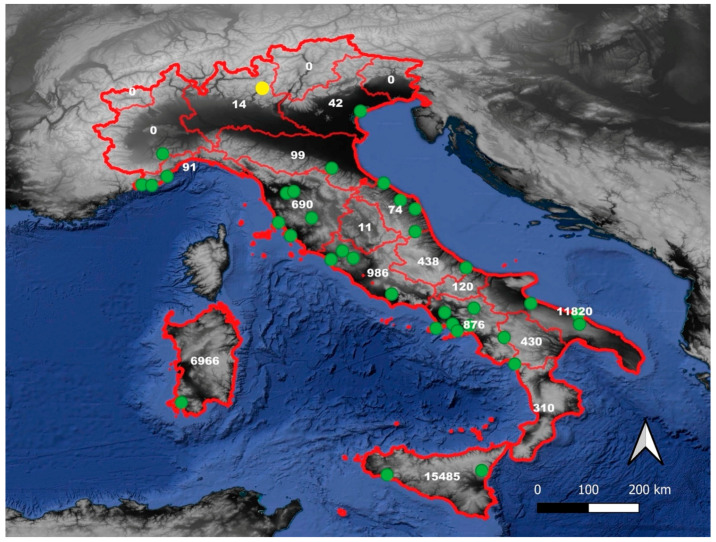
Distribution maps of artichoke landraces in Italy: “*Carciofo di Malegno*” (yellow point) and other artichoke landraces (green points). Cultivated hectares of artichoke in each region (white numbers).

**Figure 2 plants-13-00680-f002:**
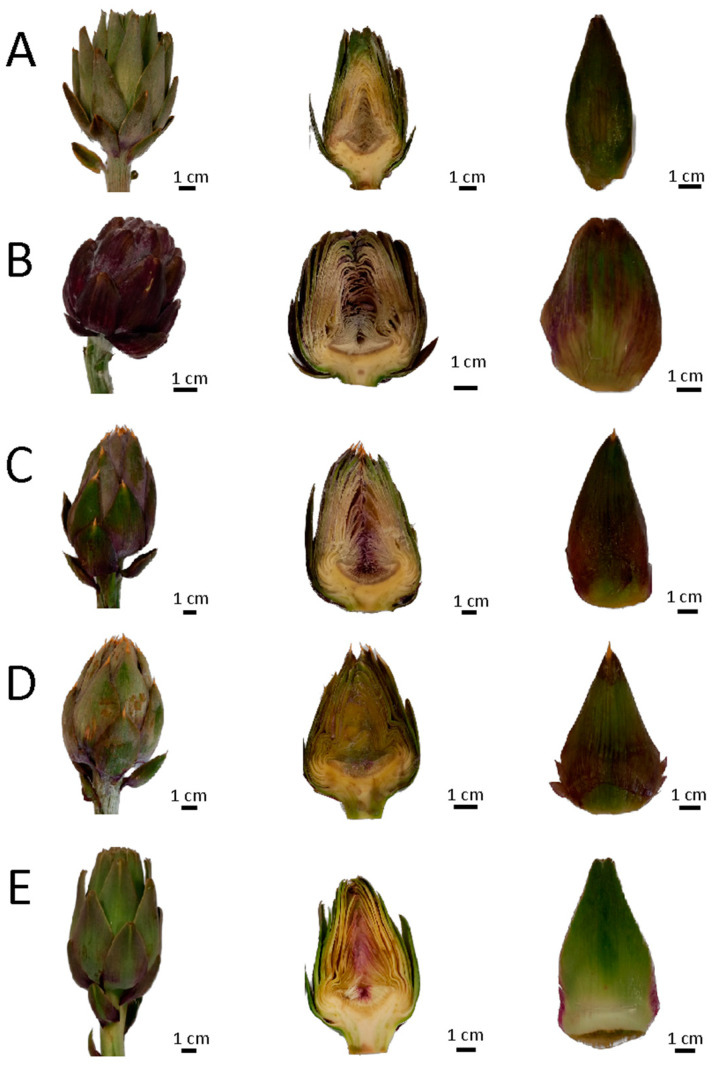
The *“Carciofo di Malegno*” (**A**) and four commercial globe artichoke (**B**–**E**).

**Figure 3 plants-13-00680-f003:**
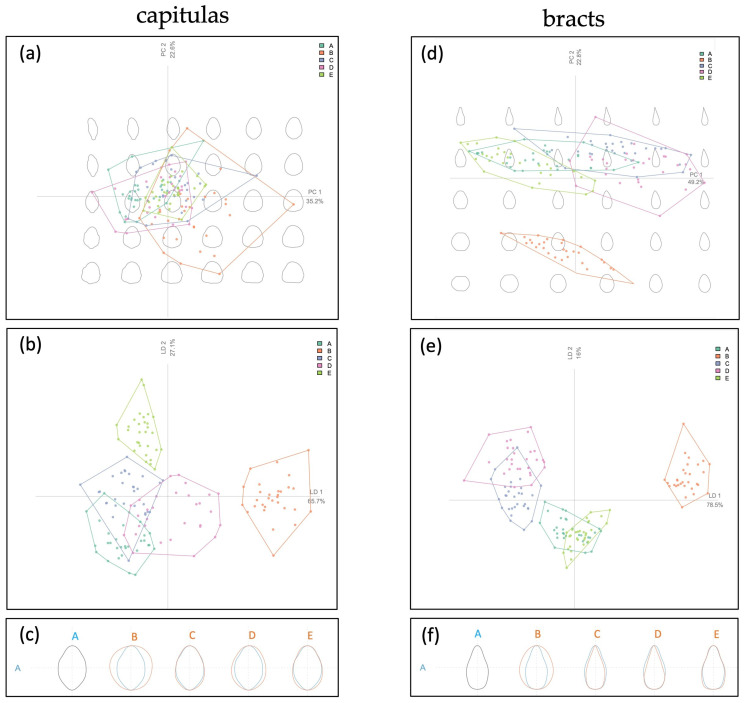
PCA biplot of artichokes’ capitula of the five genotypes (**a**); linear discriminant analysis of principal components (LDA) biplot of the artichokes’ capitula (**b**). (**c**) Mean shape of artichokes’ capitula of the five genotypes. (**d**) PCA biplot of artichokes’ bracts of the five genotypes; (**e**) linear discriminant analysis of principal components (LDA) biplot of the artichokes’ bracts. (**f**) Mean shape of artichokes’ capitula of the five genotypes. Grey figures in the background (**a**,**d**) show reconstructions of artichoke shape (capitula and bracts, respectively) according to each position in the multidimensional space. The identification code of artichoke samples (capital letter) is referred to in Figure 2.

**Figure 4 plants-13-00680-f004:**
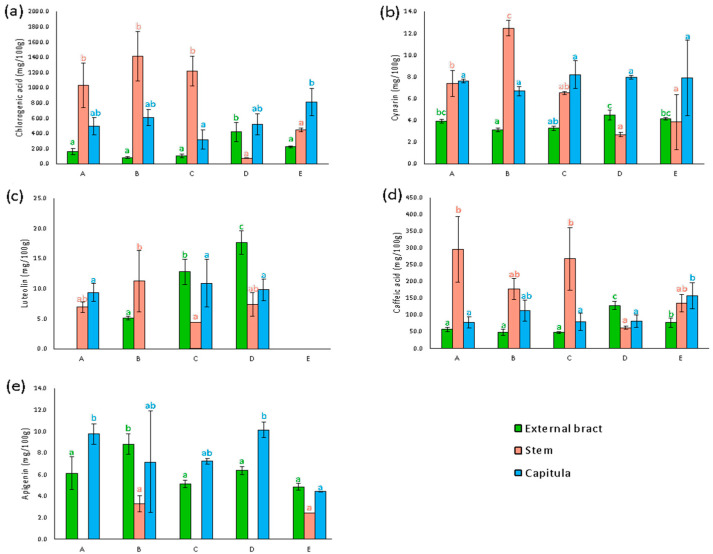
Results of phytochemical assays on external bract, steam, and capitula of artichoke samples. Chlorogenic acid (**a**), cynarin (**b**), luteolin (**c**), caffeic acid (**d**), and apigenin (**e**) content were expressed in mg/100 g dw. The identification code of artichoke samples (capital letter) is referred to in Figure 2.

**Figure 5 plants-13-00680-f005:**
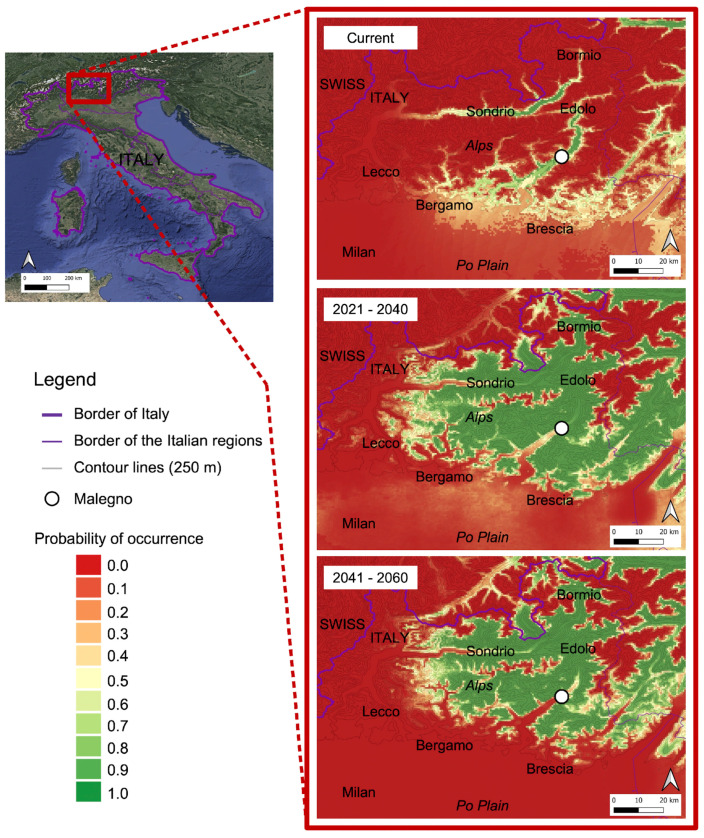
Projections of the spatial distribution of “*Carciofo di Malegno*” according to MaxEnt across the north of Italy at present with occurrence points and at horizon 2021–2040 and 2041–2060.

**Table 1 plants-13-00680-t001:** Origin, shape, color, and consumable part of artichoke sample. Stem, non-edible, and edible parameters are indicated with mean ± standard deviation. The identification codes of artichoke samples (capital letter) are referred to in Figure 2.

Code	Origin	Thorn/Mucro	Head Shape	External Color	Stem (g)	Non-Edible (g)	Edible (g)
A	Malegno (BS)	Mucro	Conical	Green	8.73 ± 0.63	22.13 ± 4.39	35.12 ± 3.94
B	Sicily	Short thorn/absent	Spherical	Violet with green tones	35.01 ± 7.33	106.19 ± 10.13	109.29 ± 15.98
C	Sicily	Thorn	Conical	Green with violet tones	36.04 ± 10.23	59. 67 ± 8.52	8.51 ± 10.28
D	Sardinia	Thorn	Conical	Green with violet tones	26.18 ± 1.05	44.65 ± 7.02	87.99 ± 10.44
E	Apulia	Short thorn/absent	Ovoidal	Green with violet tones	30.12 ± 2.49	47.07 ± 7.96	85.67 ± 5.84

**Table 2 plants-13-00680-t002:** Environmental variables used for modeling the potential distribution of *“Carciofo di Malegno*”.

Code/Unit	Bioclimatic Variable
BIO1 (°C)	Annual Mean Temperature
BIO2 (°C)	Mean Diurnal Range (Mean of monthly (max temp–min temp)
BIO3 (-)	Isothermality (BIO2/BIO7 × 100)
BIO4 (°C)	Temperature Seasonality (standard deviation × 100)
BIO5 (°C)	Max Temperature of Warmest Month
BIO6 (°C)	Min Temperature of Coldest Month
BIO7 (°C)	Temperature Annual Range (BIO5-BIO6)
BIO8 (°C)	Mean Temperature of Wettest Quarter
BIO9 (°C)	Mean Temperature of Driest Quarter
BIO10 (°C)	Mean Temperature of Warmest Quarter
BIO11 (°C)	Mean Temperature of Coldest Quarter
BIO12 (mm)	Annual Precipitation
BIO13 (mm)	Precipitation of Wettest Month
BIO14 (mm)	Precipitation of Driest Month
BIO15 (-)	Precipitation Seasonality (Coefficient of Variation)
BIO16 (mm)	Precipitation of Wettest Quarter
BIO17 (mm)	Precipitation of Driest Quarter
BIO18 (mm)	Precipitation of Warmest Quarter
BIO19 (mm)	Precipitation of Coldest Quarter

## Data Availability

Data are contained within the article and Appendix A.

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
