# Peer review of "Characterization and Future Distribution Prospects of “Carciofo di Malegno” Landrace for Its In Situ Conservation"

_plants, 2024, doi:10.3390/plants13050680_

Round 1

Reviewer 1 Report

Comments and Suggestions for Authors

1. I recommend modifying the title, for instance, as follows:

  "Characterization and Future distribution Prospects of Carciofo di Malegno for Conservation in situ"

2.  page 1, line 12 : I recommend changing 'conducted' to 'investigated’

3. The number of keywords is excessive. Please limit them to around 5.

4. The paper's significant focus is on anticipating the current and future distribution of 'Carciofo di Malegno.' However, there is no mention of this aspect in the introduction section. It is deemed necessary to incorporate relevant content addressing this in the introduction.

5. I recommend modifying the table caption as follows:

"Figure 1. Distribution Maps of Artichoke Landraces in Italy: 'Carciofo di Malegno' (yellow point) and Other Artichoke Landraces (green points). Cultivated Hectares of Artichoke in Each Region (white numbers)." or "Figure 1. Distribution Maps of Artichoke Landraces in Italy (the yellow point represents 'Carciofo di Malegno,' while the green points indicate other artichoke landraces). The white numbers in each region represent the cultivated hectares of artichoke."

6. "I would like to suggest a clearer description of the content on pages 4, specifically lines 131-137, as this section is crucial for expressing the research objectives.“

7. Pages 7, lines 243-244 may require the following modifications: “Twenty-five georeferenced points (occurrence points) were collected to assess on the spatial distribution~”

8. I believe it would be appropriate to predict the distribution on a nationwide scale, covering the entire Italy. If there are specific reasons to restrict it to the northern regions, these justifications should be provided within the manuscript.

Comments on the Quality of English Language

I request the improvement of sentence expressions to enhance clarity and ensure ease of understanding for readers.

Author Response

Thank you for your advices. We modified the main text (in red) as suggested.

1) recommend modifying the title, for instance, as follows: "Characterization and Future distribution Prospects of Carciofo di Malegno for Conservation in situ".

Done, we changed the title like your suggestion

2) page 1, line 12 : I recommend changing 'conducted' to 'investigated’

Done

3) The number of keywords is excessive. Please limit them to around 5.

The magazine allows up to a maximum of 10, we still removed one (Morphometric analyses)

4) The paper's significant focus is on anticipating the current and future distribution of 'Carciofo di Malegno.' However, there is no mention of this aspect in the introduction section. It is deemed necessary to incorporate relevant content addressing this in the introduction.

Done, we added more details. See line 129-130

5) I recommend modifying the table caption as follows: "Figure 1. Distribution Maps of Artichoke Landraces in Italy: 'Carciofo di Malegno' (yellow point) and Other Artichoke Landraces (green points). Cultivated Hectares of Artichoke in Each Region (white numbers)." or "Figure 1. Distribution Maps of Artichoke Landraces in Italy (the yellow point represents 'Carciofo di Malegno,' while the green points indicate other artichoke landraces). The white numbers in each region represent the cultivated hectares of artichoke."

Done

6) "I would like to suggest a clearer description of the content on pages 4, specifically lines 131-137, as this section is crucial for expressing the research objectives.“

Done, we added more details regarding the study aims. See line 133-134

7) Pages 7, lines 243-244 may require the following modifications: “Twenty-five georeferenced points (occurrence points) were collected to assess on the spatial distribution~”

Done

8) I believe it would be appropriate to predict the distribution on a nationwide scale, covering the entire Italy. If there are specific reasons to restrict it to the northern regions, these justifications should be provided within the manuscript.

We were not interested in prediction of “carciofi di Malegno’s” cultivation in entire Italy because this landrace is historically cultivated in mountain areas of the Alps and it’s precisely in these areas where it’s grows and it will be preserved and cultivated in the future. We added more details, see line

Reviewer 2 Report

Comments and Suggestions for Authors

This is a very interesting and well written manuscript, concerning the characterization of an artichoke of the Southern Alps, as well as a future scenario for its in situ conservation. As the authors reported, 'the four commercial globe artichokes (B, C, D and E) were collected from a local market'. For the repetition of these experiments in the future, its not clear if these are commercial varieties, thus with always the same characteristics, or populations or something else. Also, there was a general description and/or perhaps a assumption for the origin of these, while its not clear the collection date for each of them, the way of transportation and conservation etc. I think that all these should be clarified in the manuscript. In my opinion the manuscript is suitable for publication after major revision. Bellow are my specific remarks.

In my opinion, the introduction section is interesting but rather extended, especially in the first general part. 

In material and method section please be more specific about the four commercial globe artichokes. The information is too general. 

page 5. For each artichoke, 20 capitula were collected.... By which criteria the selection was made?

155 'weighted' . model company in parenthesis

164 Each part was frozen at -80 °C  by which way?

164 'kitchen mixer'  model company in parenthesis

165 'sealed container at -80 °C' give more details

table 1. ad value +- SD ? in the end

'External color' It will be more proper to use the suitable equipment for this. This is just an estimation

169-170. 'commercial variety (B, C, D and E) randomly collected from local markets'. Please give more details about these variety/ies

194-207. If this method was mentioned by others, please use the reference and mention only possible modifications

237 statistical program?

Author Response

Thank you for your advices. We modified the main text (in red) as suggested.

1) This is a very interesting and well written manuscript, concerning the characterization of an artichoke of the Southern Alps, as well as a future scenario for its in situ conservation. As the authors reported, 'the four commercial globe artichokes (B, C, D and E) were collected from a local market'. For the repetition of these experiments in the future, its not clear if these are commercial varieties, thus with always the same characteristics, or populations or something else. Also, there was a general description and/or perhaps a assumption for the origin of these, while its not clear the collection date for each of them, the way of transportation and conservation etc. In material and method section please be more specific about the four commercial globe artichokes. The information is too general. 169-170. 'commercial variety (B, C, D and E) randomly collected from local markets'. Please give more details about these variety/ies.

The commercial artichokes were choosen considering morphological aspects and different cultivation areas.  Genetic analisys were not carried out (they weren’t our goal) but their different morphological groups (line 151-156) suggest that are different genotype. Anyway we have added some more detail in the section of materials and methods,see line 151.

2) In my opinion, the introduction section is interesting but rather extended, especially in the first general part. 

Done, we cut part of the introduction, see the main text

3) page 5. For each artichoke, 20 capitula were collected.... By which criteria the selection was made?

We used 20 capitula because  it was the availability of  the “carciofo di Malegno” landrace that is produced only by a few farmer. So we started from the number of this landrace to decide the total number of each artichoke sample.

4) 155 'weighted' . model company in parenthesis

Done

5) 164 Each part was frozen at -80 °C  by which way?

Done

6) 164 'kitchen mixer'  model company in parenthesis

Done

7) 165 'sealed container at -80 °C' give more details

Done

8) table 1. ad value +- SD ? in the end

Done

9) 'External color' It will be more proper to use the suitable equipment for this. This is just an estimation

The external color was an estimation based on DUS forms redacted by the International Union for the Protection of New Varieties of Plants (UPOV); a official procedure.

10) 194-207. If this method was mentioned by others, please use the reference and mention only possible modifications

Done, see line 201

11) 237 statistical program?

Done, see line 234

Round 2

Reviewer 1 Report

Comments and Suggestions for Authors

This manuscript has been appropriately revised and improved for publication.

Reviewer 2 Report

Comments and Suggestions for Authors

The authors has incorporated in the text  all the suggestions of the reviewers and the manuscript was improved. Thus, in my opinion, is now suitable for publication in the journal